# Electrospun Polycaprolactone/Collagen Scaffolds Enhance Manipulability and Influence the Composition of Self-Assembled Extracellular Matrix

**DOI:** 10.3390/bioengineering12101077

**Published:** 2025-10-03

**Authors:** Saeed Farzamfar, Stéphane Chabaud, Julie Fradette, Yannick Rioux, Stéphane Bolduc

**Affiliations:** 1Department of Surgery, Faculty of Medicine, Université Laval, Québec, QC G1V 0A6, Canada; saeed.farzamfar@lhsc.on.ca (S.F.); julie.fradette@fmed.ulaval.ca (J.F.); 2Centre de Recherche en Organogénèse Expérimentale/LOEX, Regenerative Medicine Division, CHU de Québec-Université Laval Research Center, Québec, QC G1J 1Z4, Canada; stephane.chabaud@crchudequebec.ulaval.ca (S.C.);; 3Département de Génie Mécanique, Faculté de Sciences et de Génie, Université Laval, Québec, QC G1V 0A6, Canada

**Keywords:** electrospun scaffolds, extracellular matrix, self-assembly, polycaprolactone, collagen, mechanical properties, stromal cells, biomaterial, tissue engineering

## Abstract

Cell-mediated extracellular matrix (ECM) self-assembly provides a biologically relevant approach for developing near-physiological tissue-engineered constructs by utilizing stromal cells to secrete and assemble ECM components in the presence of ascorbic acid. Despite its unique advantages, this method often results in scaffolds with limited mechanical properties, depending on the cell type. This research aimed to enhance the mechanical properties of these constructs by culturing cells derived from various sources, including skin, bladder, urethra, vagina, and adipose tissue, on electrospun scaffolds composed of polycaprolactone and collagen (PCLCOL). The hybrid scaffolds were evaluated using various in vitro assays to assess their structural and functional properties. Results showed that different stromal cells could deposit ECM on the PCLCOL with distinct composition compared to the ECM that was self-assembled on tissue culture plates (TCP). Additionally, cells cultured on PCLCOL exhibited a different growth factor secretion profile compared to those on TCP. Mechanical testing demonstrated that the hybrid scaffolds exhibited high mechanical properties and superior manipulability. These findings suggest that PCLCOL could be a promising platform for developing biomimetic scaffolds that combine enhanced mechanical strength with integrated biological cues for tissue repair.

## 1. Introduction

Scaffolds derived from extracellular matrix (ECM), whether decellularized or cell-secreted, play a pivotal role in tissue engineering by providing a biomimetic environment that supports cellular activities and tissue regeneration [1,2]. In this context, cell-derived ECM self-assembly is a unique approach for producing near-physiological tissue-engineered constructs by harnessing stromal cells’ ability to secrete and assemble ECM components in the presence of ascorbic acid [3,4]. Beyond ECM production, these cells also influence epithelial cell behavior through paracrine signaling. In both normal development and tissue regeneration processes, stromal cells release a variety of growth factors and cytokines that stimulate epithelial cell growth, specialization, and the maintenance of their features [5,6]. For example, in the intestinal niche, stromal cells regulate epithelial stem cell differentiation into absorptive enterocytes, goblet cells, and Paneth cells [7]. Disruption of this communication between stromal and epithelial cells can lead to non-functional epithelial cell differentiation. It is worth noting that stromal cells within each tissue exhibit a distinct secretory profile that is crucial for the development of specialized epithelial tissue within that particular organ [8,9]. In this regard, Carrier et al. investigated how the tissue origin of stromal fibroblasts and epithelial cells influenced corneal reconstruction in vitro. Using self-assembled human tissue models, they combined corneal or skin-derived fibroblasts and epithelial cells in various pairings. They found that the differentiation, epithelial thickness, and transparency of the reconstructed tissues were significantly affected by the organ-specific origin of the cells [10]. It is important to emphasize the distinction between mesenchymal stem/stromal cells (MSCs) and fibroblasts. MSCs are multipotent progenitor cells capable of differentiating into multiple mesenchymal lineages and are characterized by a broad secretory profile that supports epithelial growth, repair, and immune modulation [11]. In contrast, fibroblasts represent a more lineage-committed stromal cell population, primarily responsible for producing and remodeling ECM within their tissue of origin [12].

Tissue-specific stromal cells are essential for developing certain types of tissue-engineered epithelialized tissues, and the ECM self-assembly method has proven to be beneficial for achieving this goal. However, a major limitation of self-assembled ECM constructs is their suboptimal mechanical strength for certain cell types and applications, which could restrict clinical translation. On the other hand, electrospun synthetic scaffolds produced from polycaprolactone (PCL) and collagen (PCLCOL scaffolds) offer outstanding mechanical properties and processability for tissue engineering applications [13,14]. These scaffolds combine the strength and stability of PCL with the biological compatibility and cell-interaction properties of collagen. Furthermore, the microarchitecture of these scaffolds promotes cell adhesion, proliferation, and ECM secretion [15,16]. However, these constructs do not possess the inherent biological cues that are present in the self-assembled ECM.

In this study, we aimed to integrate the mechanical advantages of PCLCOL with the biological specificity of stromal cell-derived ECM. We cultured human-derived fibroblasts from dermis (DFs), bladder (BFs), urethra (UFs), vagina (VFs), and adipose-derived stem/stromal cells (ASCs) on PCLCOL and compared their ECM composition and growth factor secretion profiles to those cultured on conventional tissue culture plates (TCP). Mechanical properties were also assessed. This approach seeks to develop hybrid constructs with enhanced mechanical integrity and tissue-specific biological functionality.

## 2. Materials and Methods

### 2.1. Fabrication of PCLCOL Samples Using Electrospinning

A 15% (*w*/*v*) solution of PCL (Mn 80000, Sigma Aldrich, California, CA, USA) was prepared by dissolving PCL in pure acetic acid (Merck, Darmstadt, Germany). Separately, a 17.5% (*w*/*w*) solution of collagen (isolated from rat tail) was prepared in a mixture of acetic acid and non-pyrogenic water (1:1). The collagen solution was then added dropwise to the PCL solution and stirred on a high-speed magnetic stirrer for 4 h. The polymeric solution containing 20% collagen and 80% PCL by weight was stored at −80 °C until further use. The solution was later used to fabricate electrospun mats using an electrospinning device with a feeding rate of 0.2 mL/h, a positive voltage of 15 kV, a needle-to-collector distance of 15 cm, and a mandrel rotation speed of approximately 300 rpm.

### 2.2. Microstructure Analysis

The structure of PCLCOL was assessed through scanning electron microscopy (SEM). The scaffolds were coated with gold-palladium and observed under SEM at 20 kV using Tescan Vega II (Brno, Czech Republic). Fiber size measurement was conducted by randomly selecting fibers and measuring their diameters using ImageJ software (NIH, Bethesda, MD, USA, Version V1.54g).

### 2.3. Mechanical Strength Analysis

The mechanical characteristics of both cell-free and hybrid scaffolds (scaffolds coated with cell-secreted self-assembled ECM) were evaluated through uniaxial tensile testing using an ElectroPuls E1000 mechanical tester (Instron, Norwood, MA, USA) as described before [17]. Bone-shaped samples were obtained using a custom stainless-steel punch. Each sample was stretched at a constant rate of 0.1 mm/s until rupture occurred. The maximum strength (N) required to tear the constructs was recorded and the ultimate tensile strength (UTS) and elastic modulus were determined using the stress–strain curves. The overall tissue thickness was determined from Masson’s trichrome stained histological cross-sections using ImageJ software (NIH, Bethesda, MD, USA).

The mechanical properties of the self-assembled ECM produced on the TCP were assessed using a biaxial puncture test installed on the ElectroPuls E1000 mechanical tester (Instron, Norwood, MA, USA). Samples were mounted between two cylindrical discs with a circular opening and a rubber ring on the gripping surfaces. A 2-mm spherical probe, mounted on the indenter head, was used to puncture the samples at a speed of 20 μm/s. Three samples were used for each experiment.

### 2.4. Fourier-Transform Infrared Spectroscopy (FTIR) Assay

The chemical composition of the scaffolds was analyzed using a Fourier-transform infrared-ATR (FTIR-ATR) spectrometer (Jasco 4600, Hachioji, Tokyo, Japan). Spectra were recorded in the range of 600–4000 cm^−1^ at room temperature, using an average of 64 scans per sample and a resolution of 4 cm^−1^.

### 2.5. Degradation Rate Measurement

The dry scaffolds were weighted and sterilized in 70% *v*/*v* ethanol for 1 h, rinsed in Phosphate-Buffered Saline (PBS) and then immersed in Dulbecco-Vogt modification of Eagle’s medium (DMEM, Invitrogen, Burlington, ON, Canada) containing 10% Fetal Bovine Serum (FBS, Avantor Seradigm FB Essence, Randor, PA, USA), 50 µg/mL of 2-phospho-l-ascorbic acid (Sigma Aldrich, California, CA, USA), and 1% antibiotics (Penicillin and Streptomycin, Sigma Aldrich, California, CA, USA) and kept for 60 days at 37 °C. Every ten days, scaffolds were removed from the media and lyophilized for 48 h. Finally, the following formula was used to calculate the degradation rate at each time step.$$\mathrm{Degradation}\;\mathrm{rate}\;(\%)=(\frac{W0-W1}{W0})\;\times \;100$$where W0 denotes the dry weight of scaffolds and W1 represents the weight of scaffolds after the lyophilization process.

### 2.6. Cell Culture

DFs, BFs, UFs, and VFs, and ASCs were obtained from healthy human donors undergoing surgery for a non-oncologic condition and utilized for cell culture studies. These cell populations have been previously established and characterized in our laboratory, as described in earlier studies [18,19,20,21]. Cells were used at passages 3 to 5 and cultured in DMEM (Invitrogen, Burlington, VT, Canada) containing 10% FBS, 100 U/mL penicillin, and 25 mg/mL gentamicin. For the self-assembled ECM production, complete culture media was supplemented with 50 µg/mL of 2-phospho-L-ascorbic acid (Sigma Aldrich, California, CA, USA). Only one biological replicate was used for cell extraction. The cell culture was performed for all cell types in the same incubator simultaneously. Table 1 summarizes the characteristics of the human cells used in the study.

### 2.7. Cell Adhesion Assay

PCLCOL were cut into circular pieces and immersed in 70% *v*/*v* ethanol for 1 h. Then, they were allowed to dry under the laminar flow hood for 2 h. They were then rinsed 6–8 times with sterile phosphate-buffered saline (PBS) solution. Subsequently, the scaffolds were placed into the wells of a 6-well plate (Falcon, Thermo Fisher, Waltham, MA, USA), secured in place with custom-made metal rings, and incubated with DMEM supplemented with 20% *v*/*v* FBS, and 1% antibiotics for 24 h. Afterward, the media in the wells was discarded, and the scaffolds were seeded with stromal cells at a density of 40,000 cells per cm^2^ in 100 µL of complete media. Following seeding, the cells were incubated for 3 h. Then, the cell-scaffold constructs were removed and slowly rinsed with complete media. The wash media was collected, and the number of cells in it, corresponding to the number of detached cells from the scaffolds, was counted using the Beckman Coulter Cell Counter (California, CA, USA). Finally, the following formula was used to calculate the percentage of cell adhesion to the scaffolds (indirect assessment):$$\mathrm{Cell}\;\mathrm{adhesion}\;(\%)\;=\;(\frac{N0-N1}{N0})\;\times \;100$$
where N0 is the number of the cells seeded onto the scaffolds and N1 is the number of cells in the wash media.

### 2.8. Cell Viability Assay

Scaffolds were cut, sterilized and prepared for cell seeding according to the protocol as described in Section 2.7. Stromal cells were seeded onto the scaffolds at the density of 10,000 cells per scaffold in 100 µL of media and incubated for 3 h. Then, complete culture media supplemented with 2-Phospho-l-ascorbic acid (50 µg/mL) was slowly added into each well and cells were cultured for 10 days. On days 3, 5, and 10, a cell viability assay was performed using PrestoBlue™ Cell Viability Reagent (Invitrogen, Burlington, VT, Canada). Briefly, scaffolds were removed from the wells and immersed in the 500 µL of PrestoBlue™ Cell Viability Reagent prepared in the complete culture media at 10 *v*/*v*%. The samples were then incubated for 2 h. Subsequently, 100 µL of the solution was transferred into 96-well plates and their absorbance was measured at 570 nm using a plate reader (SPECTRAmax PLUS, Molecular Devices, California, CA, USA). The stromal cells seeded onto the TCP without scaffolds were used as the control group, and the relative cell viability was calculated based on the optical density of this group.

### 2.9. Production of the Self-Assembled ECM on the PCLCOL

Scaffolds were cut, sterilized, and prepared for cell seeding following the protocol outlined in Section 2.7. Stromal cells were seeded onto the scaffolds at a density of 40,000 cells per cm^2^ in 100 µL of media and incubated for 3 h. Subsequently, complete culture media supplemented with ascorbate at a concentration of 50 µg/mL was gently introduced into each well, and the cells were cultured for a period of 14 days. Following this initial culture period, a second round of cell seeding was conducted. The culture media in each well was removed, and the cells were seeded onto the scaffolds at the same density of 40,000 cells per cm^2^ in 100 µL of media. After the 3-h incubation period, complete culture medium supplemented with ascorbate at 50 µg/mL was slowly added into each well. The cells were then cultured for an additional 14 days.

### 2.10. Production of Self-Assembled ECM on TCP

Stromal cells were seeded to achieve a final density of 40,000 cells per cm^2^ in 6-well plates (Falcon, Thermo Fisher, Waltham, MA, USA) and cultured in complete medium supplemented with 50 µg/mL of ascorbate for a duration of 14 days. On the 14th day, a second cell seeding was conducted on the newly formed stromal sheets. The culture was continued until the 28th day to allow for sufficient neosynthesized ECM, resulting in the formation of manipulable cellularized sheets [22].

### 2.11. Histological Analysis

The developed tissues were fixed in 3.7% formaldehyde, embedded in paraffin, and sectioned at thickness of 5 µm. The slides were then stained with Masson’s trichrome (MT). MT-stained tissue slides were imaged using a Zeiss Axio Imager M2 microscope equipped with an AxioCam ICc1 camera (Oberkochen, Germany). Thickness measurements were made using Image J software.

### 2.12. Dot Blot Assay

ECM extracts were prepared using a BRANSON Digital Sonifier 450 ultrasonic processor (Branson Ultrasonics Corporation, Connecticut, CT, USA). Briefly, the ECM samples were collected and resuspended in cold phosphate-buffered saline (PBS). The suspension was then sonicated on ice using a 3 mm probe with 3-s on/off cycles for a total of 2 min. The resulting homogenates were centrifuged at 10,000× *g* for 20 min at 4 °C, and the supernatants were collected for protein quantification and subsequent dot blot analysis. For the dot blot analysis, 50 μg of ECM proteins produced via self-assembly method on the PCLCOL and TCP were loaded into each well and transferred onto a PVDF membrane (BIORAD, California, CA, USA) using a Bio-Dot Apparatus (Bio-Rad, Mississauga, ON, Canada). Rehydration of the membrane was achieved by injecting 200 μL of a Tris-buffered solution (TBS) into each well. Following this, the antigenic sites were blocked for 1 h in TBS-T containing a 5% powdered milk solution (Non-Fat Powdered Milk, Bio Basic, Markham, ON, Canada) and 0.1% Tween-20. Subsequently, the membranes were incubated for 24 h with primary antibodies at 4 °C (Table 2). Then, the membranes were washed six times with PBS containing 0.1% Tween-20: three quick washes followed by three slow washes on a shaker, with each round lasting 3–5 min. Then, the membranes were incubated with secondary antibodies (Table 2) for 2 h at room temperature under gentle shaking. Detection of the proteins of interest was accomplished using a SuperSignal™ West Dura Extended Duration Substrate (Thermo Fisher Scientific, Waltham, MA, USA) and the Fusion Fx7 imager (Vilbert-Lourmat, Marne-La-Vallée, France). Quantification of the dot blots was conducted through densitometry utilizing ImageJ (NIH, Bethesda, MD, USA). Protein expression was normalized by the surface area of the cell culture substrate. Relative protein expression was calculated using the protein expression of DFs cultured on the TCP.

### 2.13. Growth Factors Secretion Profile Analysis

The scaffolds were cut, sterilized, and prepared for cell seeding according to the protocol described in Section 2.7. Stromal cells were seeded onto the scaffolds at a density of 40,000 cells per cm^2^ in 100 µL of media and incubated for 3 h. Following this, complete culture media, supplemented with ascorbate at a concentration of 50 µg/mL, was gently added to each well, and the cells were cultured for 10 days. Stromal cells were also seeded at a final density of 40,000 cells per cm^2^ in 6-well plates and maintained in complete medium supplemented with 50 µg/mL of ascorbate for 10 days. The culture media was changed three times a week. On day 10, the culture media from each well were collected and stored at −80 °C. Then, the expression levels of various growth factors in the supernatant of stromal cells cultured on PCLCOL were compared with those in the supernatant of the same cells cultured under the same conditions on the TCP using The RayBio C-Series Human Growth Factor Antibody Array C1 Kit (RayBiotech, Inc., Georgia, GA, USA). A total of 41 targets were detected following the instructions provided by the manufacturer. Signals from membranes were detected using a Fusion Fx7 (Vilbert-Lourmat, Marne-La-Vallée, France), and subsequently analyzed using ImageJ software (NIH, Bethesda, MD, USA). Relative protein expression was calculated based on the growth factors expression in the supernatant of the DFs cultured on the TCP.

### 2.14. Statistical Analysis

Data were analyzed using GraphPad Prism version 5 via one-way ANOVA and Student’s *t*-test. All data were expressed as mean ± standard deviation, and experiments were repeated at least three times.

## 3. Results

### 3.1. Microstructure Analysis Revealed Fibrous Structure of the Electrospun Scaffolds

The results indicated that the PCLCOL consisted of randomly arranged fibers with a smooth surface (Figure 1). No bead formation was detected, and the fibers showed no signs of disintegration. Mean fiber size measurement was 264.34 ± 94.06 nm.

### 3.2. Confirmation of Collagen Incorporation into PCL Using FTIR Spectroscopy

The FTIR spectra (Figure 2) of the PCLCOL exhibited characteristic peaks corresponding to both PCL and collagen components. The spectra showed the amide I and amide II bands of collagen, which were observed around 1650 cm^−1^ and 1550 cm^−1^, respectively. The presence of these specific peaks confirmed the successful incorporation of collagen into the PCL matrix. Table 3 summarizes the important peaks observed in the FTIR assay.

### 3.3. Time-Dependent Degradation Behavior of PCLCOL in Culture Media

Degradation test results (Figure 3) showed that PCLCOL slowly degraded while being incubated at 37 °C in the culture media. The rate of degradation increased over time and reached 9.84 ± 0.696% by the end of the 60th day of immersion in the culture media.

### 3.4. PCLCOL Supports Strong and Consistent Adhesion Across Stromal Cell Types

The percentage of cell adhesion (Figure 4) for DFs, BFs, UFs, VFs, and ASCs to the PCLCOL was 84.0 ± 3.2, 78.4 ± 3.6%, 82.4 ± 4.4%, 85.7 ± 3.8%, and 79.3 ± 8.8%, respectively. Differences between MSCs type were not statistically significant.

### 3.5. Initial Decline and Subsequent Stabilization of Stromal Cell’s Viability on PCLCOL

The cell viability assay results (Figure 5) indicated that on day 3, the relative viability of DFs, BFs, and ASCs grown on the scaffolds was significantly reduced compared to cells cultured on TCP. Meanwhile, there were no statistically significant differences in the relative viability of UFs and VFs between the scaffold and TCP groups at the same time point. On day 5, all stromal cell types exhibited significantly lower relative cell viability on the scaffolds than on the TCP. But by day 10, the relative cell viability of the stromal cells from all sources stabilized and was not significantly different between the PCLCOL and TCP groups.

### 3.6. PCLCOL Promotes Superior ECM Thickness in UFs, VFs, and ASCs Compared to TCP

All stromal cells cultured on the PCLCOL produced a dense layer of stromal tissue (Figure 6), although some holes and defects were observed among the ECM fibers. The cells were heterogeneously distributed within the deposited ECM. MT staining images of the ECM produced by stromal cells cultured on the TCP also revealed dense and well-organized collagen fibers, with a heterogeneous distribution of cells within the matrix. Comparing the ECM thickness between groups, we found that UFs, VFs, and ASCs cultured on PCLCOL produced significantly thicker ECM than the same cells cultured on TCP. However, the thickness of ECM produced by DFs and BFs cultured on TCP and PCLCOL was not significantly different.

### 3.7. Stromal Cell-Specific ECM Protein Secretion Profiles on TCP and PCLCOL

Results (Figure 7) showed that different stromal cells exhibited distinct ECM component secretion profiles on the TCP and PCLCOL. UFs and ASCs cultured on the TCP secreted significantly higher amounts of type I collagen compared to the same cells cultured on PCLCOL. However, the differences in type I collagen secretion between DFs, BFs, and VFs cultured on TCP and PCLCOL were not statistically significant. BFs and VFs cultured on the PCLCOL secreted significantly higher amounts of type III collagen compared to the same cells cultured on TCP. In contrast, ASCs cultured on TCP had significantly higher levels of type III collagen secretion than those cultured on PCLCOL. The expression levels of elastin for ASCs cultured on PCLCOL was significantly higher than ASCs cultured on TCP. The expression levels of fibrillin in all cell types cultured on the TCP and PCLCOL did not differ significantly.

Quantification of dot blot signals (Figure 8) for fibronectin, laminin, tenascin C, tenascin X, and thrombospondin-2 showed that among all comparisons, a statistically significant difference was observed only for fibronectin. In fact, UFs cultured on TCP secreted significantly higher levels of fibronectin compared to their counterparts on PCLCOL. No other significant differences in protein expression were found between the TCP and scaffold conditions for any of the other cell types or matrix proteins evaluated.

### 3.8. Stromal Cells Cultured on PCLCOL and TCP Exhibited Distinct, Substrate-Dependent Growth Factor Secretion Profiles

Different stromal cells cultured on TCP and PCLCOL exhibited distinct growth factors secretion profiles (Figure 9). A comprehensive analysis identified a total of 41 targets across all samples. Among the detected targets, Hepatocyte Growth Factor (HGF) is biologically active and relevant to tissue repair [23], and it was consistently and strongly overexpressed across nearly all groups. This high baseline abundance introduced bias by masking the relative differences in secretion profiles of other growth factors. For this reason, HGF was excluded from the quantitative analysis. The remaining 40 targets were analyzed to assess the differences in secretion profiles between the stromal cells cultured on TCP and those on PCLCOL. This analysis revealed distinct patterns of cytokine and growth factor secretion, indicating that the substrate material and cell source significantly impact the cellular secretion behavior.

### 3.9. Stromal Cell Seeding Did Not Alter PCLCOL Scaffold Mechanics, but ECM Constructs on TCP Showed Cell-Dependent Differences

The ultimate tensile strength (UTS), elastic modulus, εmax, and maximum force of the hybrid scaffolds did not differ significantly from those of the bare electrospun PCLCOL constructs (Figure 10). Additionally, differences between the various stromal cell groups were not statistically significant.

Assessing the mechanical properties of the self-assembled tissues produced on the TCP, we found that UFs had significantly higher maximum force and stiffness than the ASCs group. In addition, DFs group had significantly higher toughness than the ASCs group (Figure 11).

## 4. Discussion

Electrospun scaffolds, designed to replicate the microarchitecture of ECM, are gaining momentum in tissue engineering due to their capacity to enhance cellular functions and support tissue regeneration [24,25]. Their versatility allows for the creation of scaffolds with adjustable properties, making them interesting models for studying cell behavior [26]. In this study, we used electrospun PCLCOL as a platform for developing cell-secreted self-assembled ECM tissues. The novelty of this study lies in its systematic evaluation of tissue-specific stromal cells cultured on PCLCOL and the demonstration of how this platform distinctly modulates both ECM composition and growth factor secretion profiles compared to conventional self-assembly on TCP. The composition and structural characteristics of these scaffolds can be tuned to influence various cellular behaviors, including adhesion, proliferation, and growth factor/ECM secretion profiles [27,28]. Delaine-Smith et al. demonstrated that mature osteoblasts or osteogenic mesenchymal progenitor cells deposited matrix in an isotropic manner on random electrospun scaffolds. In contrast, aligned fibrous constructs exhibited a higher degree of anisotropy [29]. Cells detect the scaffold’s characteristics, like stiffness and topography, through mechanosensitive receptors. These proteins convey mechanical signals to the cell and initiate biochemical responses that influence cell behavior [30,31]. In addition, cells perceive their microenvironment characteristics through focal adhesion molecules and signaling mechanisms that are still being explored [32]. Interactions between the ECM and integrins regulate cell behavior through the focal adhesion kinase-Src and RhoA/ROCK signaling pathways [33]. However, cell interactions with PCL-only scaffolds are generally poor due to their surface hydrophobicity; incorporating collagen into the PCL structure enhances the scaffold’s surface hydrophilicity and promotes cell adhesion and proliferation [14,34]. The FTIR spectra confirmed successful collagen incorporation into the PCL scaffolds, with distinct peaks for PCL and collagen. Collagen features arginine-glycine-aspartic acid (RGD) motifs that are recognized by cell surface integrins, enabling cells to bind to the collagen-containing surface, enhancing adhesion and spreading [35,36].

Biodegradability is crucial in regenerative medicine and tissue integration. PCLCOL scaffolds degrade via hydrolysis and enzymatic action, releasing caproic acid, potentially causing local acidification and mild inflammatory responses if unbuffered. However, the body’s natural buffering systems help neutralize the localized acidity, minimizing adverse effects [37,38,39].

Despite most cells attaching to the scaffolds, an intriguing observation in this study is the transient decline in stromal cell viability on PCLCOL scaffolds during the early culture period (days 3–5). While viability recovered by day 10, the initial reduction raises questions about the underlying mechanisms. The reduced viability observed in early culture may reflect a transient dormancy phase, as stromal cells adapt to the altered mechanical and topographical cues of the scaffold. When stromal cells are cultured on scaffolds, they might enter a state of dormancy due to the distinct biophysical signals from the new microenvironment [40]. Stromal cell dormancy is characterized by a state of reduced metabolic activity. It seems that the stiffness of scaffold material has a major impact on the stromal cells’ metabolic activity [41,42,43]. However, further research is needed to determine whether the early decline in viability on PCLCOL scaffolds is linked to apoptosis, altered proliferation dynamics, or metabolic dormancy, and to assess how these factors may confound subsequent ECM deposition and growth factor secretion.

Histological evaluation showed that stromal cells cultured on the PCLCOL were able to deposit ECM, like the cells cultured on TCP. Dot blot assay results further showed that some ECM components were differentially expressed between the cells cultured on TCP and PCLCOL. Despite evidence indicating that the ECM deposition process depends on functional core secretion machinery, the molecular mechanisms through which stromal cells control ECM secretion are still not well understood [43]. Clément et al. demonstrated that the DFs-derived exosomes, when cultured in a 2D cellular monolayer versus in self-assembled tissues, contained notably different quantities of proteins associated with the formation, organization, and remodeling of the ECM [44]. Cells interact with their surrounding ECM through a process called dynamic reciprocity, where they constantly alter their microenvironment both biochemically and mechanically. This interaction includes the production and breakdown of the matrix, which continuously adjusts the cellular environment. This two-way interaction influences signaling pathways, gene expression, and cellular behavior, allowing cells to adapt to environmental changes [45].

The growth factor secretion profiles of different stromal cells varied significantly when cultured on the TCP compared to the PCLCOL. Cells adjust their growth factor secretion based on the culture substrate through various mechanisms. The topography of a surface and how cells adapt to it can greatly impact the secretion abilities of MSCs. Various surface structures can alter cell shape and morphology, which then influences the release of specific factors [9]. Yang et al. demonstrated that bone marrow-derived MSCs cultured in three-dimensional scaffolds had enhanced mRNA expression and protein production of soluble immune-related factors compared to traditional two-dimensional monolayer cultures [46].

The hybrid scaffolds exhibited superior manipulability for tensile testing and they showed a high tensile strength in the mechanical testing. In contrast, the self-assembled ECM sheets were extremely fragile and prone to wrinkling or tearing during handling, making their reliable mounting onto a tensile testing device unfeasible. The better manipulability of hybrid scaffolds can be attributed to the presence of PCL in their structure. The crystalline regions in the PCL provide the material with significant strength and stiffness as these organized chains resist deformation under stress [47]. In contrast, the amorphous regions, where the chains are more randomly oriented and loosely packed, contribute to PCL’s flexibility and ductility, allowing it to deform without breaking [48].

Differences in the mechanical properties of tissues produced by different stromal cells on TCP could be attributed to several factors. In fact, stromal cells secrete distinct types and amounts of ECM proteins, such as collagen, elastin, fibronectin, and proteoglycans, which determine the ECM’s mechanical properties. This is in accordance with the results of dot blot assay. The metabolic activity of stromal cells also varies, affecting the rate at which they produce and remodel the ECM [49,50]. Additionally, the structural organization of ECM components differs, with some cells generating more aligned ECM that enhance tensile strength in specific directions, while others produce a more random network [51,52,53].

A key limitation of this study is that only one donor was used per stromal cell type, which constrains the generalizability of the findings. It is well established that mesenchymal cells exhibit marked donor-to-donor variability in their secretory behavior, proliferation capacity, and ECM deposition potential, influenced by factors such as age, sex, and health status of the donor [54,55]. Consequently, the differences in ECM composition and growth factor secretion observed here may in part reflect donor-specific characteristics rather than universal features of the tissue of origin.

In summary, this study demonstrates that electrospun PCLCOL provide a mechanically robust and biologically responsive platform for stromal cell culture. These scaffolds support tissue-specific ECM deposition and modulate growth factor secretion, offering a promising strategy for engineering functional epithelialized tissues in future work. Future work will focus on integrating epithelial layers seeded on the hybrid scaffolds and evaluating the in vivo performance of these constructs to advance their clinical translation.

## 5. Conclusions

This study shows that electrospun PCLCOL not only provide a biomimetic structure that supports stromal cell attachment and ECM deposition but also significantly enhance the manipulability of self-assembled tissues. The hybrid PCLCOL yielded constructs with high mechanical strength. Biochemical analyses further revealed that stromal cells cultured on these scaffolds exhibited distinct ECM compositions and growth factor secretion profiles, highlighting the profound influence of substrate properties on cellular behavior and secretory function. Collectively, these findings underscore the potential of PCLCOL as a robust platform for engineering mechanically resilient and biologically active tissue constructs for regenerative medicine applications. Future studies will focus on directly assessing the functional consequences of the altered ECM composition and growth factor secretion, particularly their impact on epithelialization and tissue repair, through epithelial co-culture models and in vivo validation.

## Figures and Tables

**Figure 1 bioengineering-12-01077-f001:**
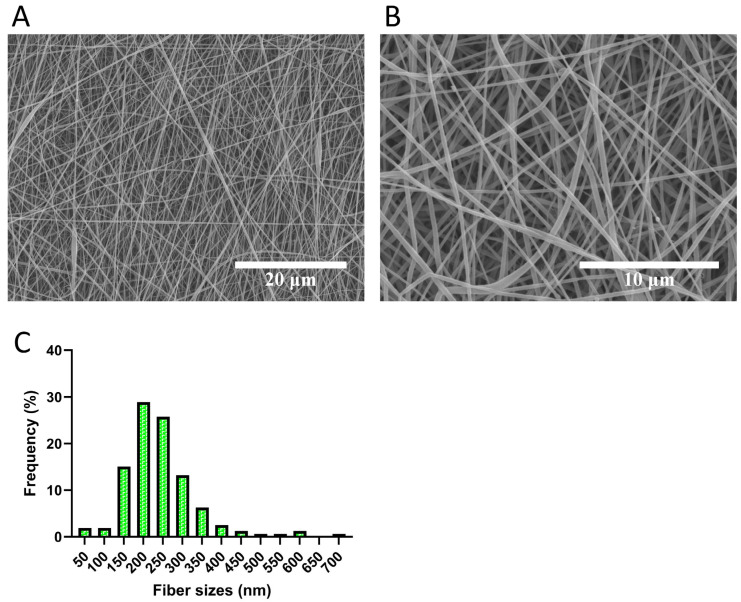
Representative SEM images of electrospun PCLCOL scaffolds. (**A**,**B**) show low- and high-magnification images showing randomly oriented, bead-free fibers with smooth surfaces. (**C**) shows fiber diameter distribution of PCLCOL scaffolds. Frequency (%) is calculated based on the proportion of fibers falling within a specific size range relative to the total number of fibers measured.

**Figure 2 bioengineering-12-01077-f002:**
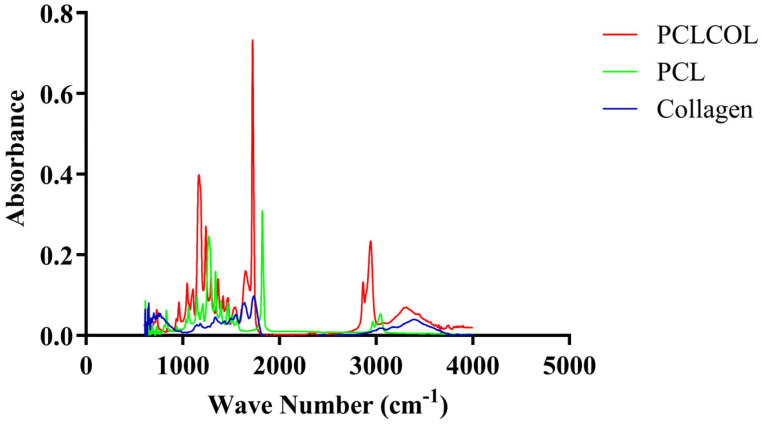
FTIR spectra of PCL, collagen, and PCLCOL.

**Figure 3 bioengineering-12-01077-f003:**
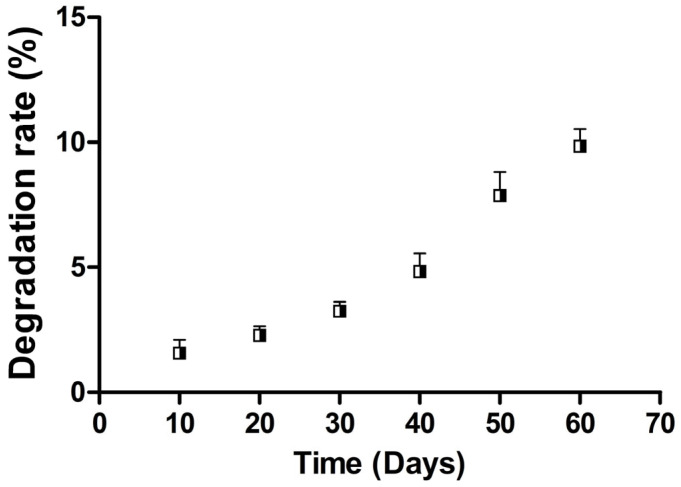
Degradation rate of PCLCOL during the 60-day incubation in DMEM culture media containing 10% FBS, 1% antibiotics, and 50 µg/mL of 2-phospho-L-ascorbic acid at 37 °C. Each experiment was repeated three times.

**Figure 4 bioengineering-12-01077-f004:**
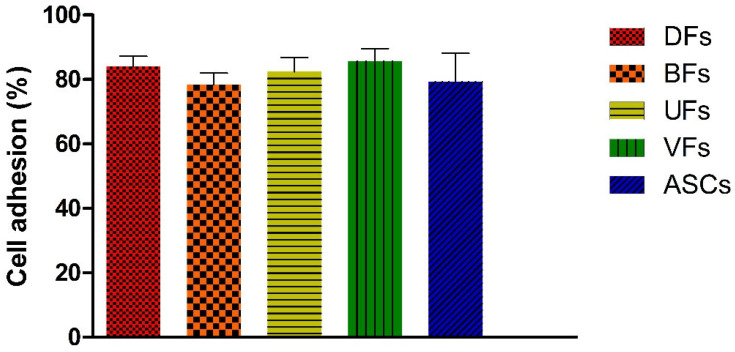
Histograms comparing the percentage of stromal cells adhesion to the PCLCOL. Each experiment was repeated three times.

**Figure 5 bioengineering-12-01077-f005:**
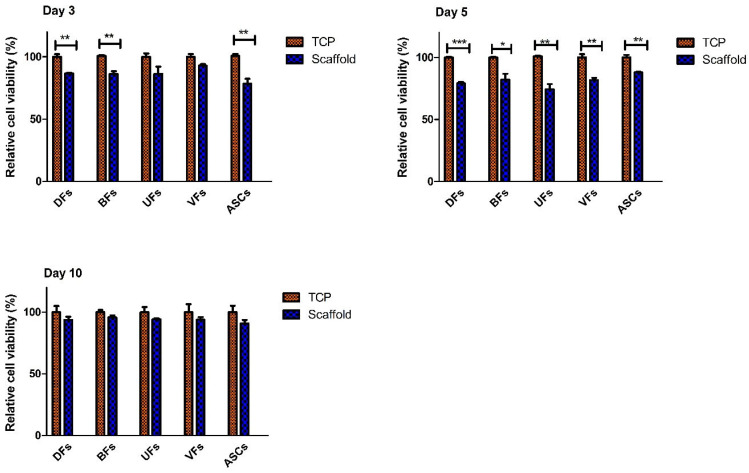
Histograms comparing the relative cell viability of different stromal cells cultured on the PCLCOL compared with the same cells cultured using the same conditions on TCP. * *p*-value < 0.05, ** *p*-value < 0.01, and *** *p*-value < 0.001. Each experiment was repeated three times.

**Figure 6 bioengineering-12-01077-f006:**
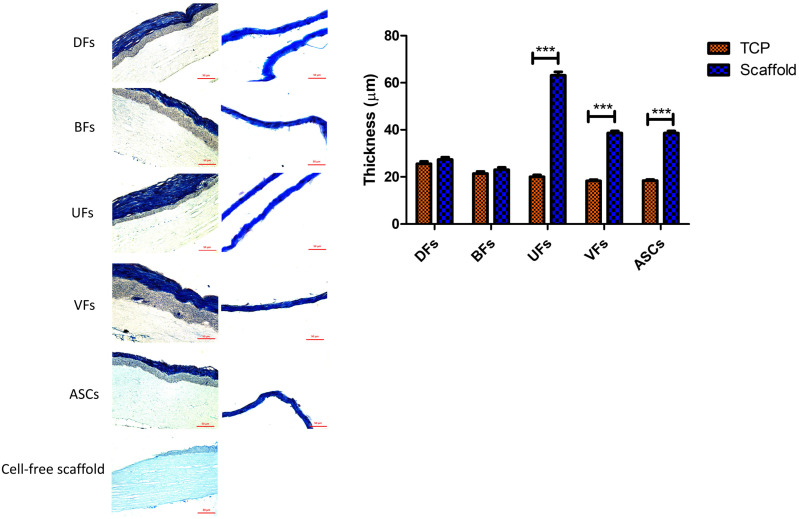
Histological evaluation and quantification of ECM thickness produced by different stromal cells cultured on TCP and PCLCOL. Representative Masson’s trichrome (MT) staining images show ECM morphology for DFs, BFs, UFs, VFs, and ASCs on both TCP (right panels) and PCLCOL (left panels). Bar graph quantifies ECM thickness, showing significantly increased thickness in UFs, VFs, and ASCs on PCLCOL compared to TCP (*** *p* < 0.001). Each experiment was repeated three times.

**Figure 7 bioengineering-12-01077-f007:**
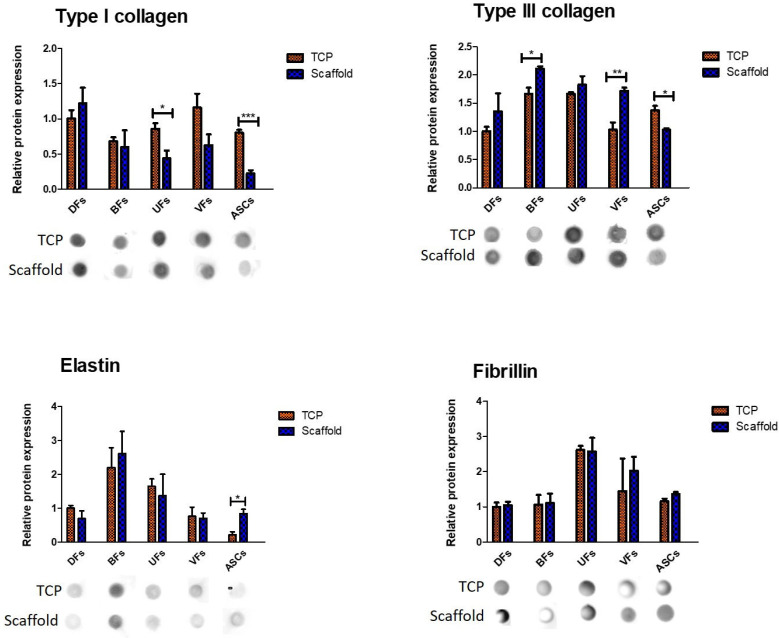
Relative expression of ECM proteins by different stromal cells cultured on PCLCOL versus TCP, assessed via dot blot analysis. Signals were detected using chemiluminescent substrate and imaged with the Fusion Fx7 system. Dot intensities were quantified using ImageJ, normalized by the culture surface area, and expressed relative to DFs cultured on TCP. Bar graphs show mean ± SD, with significant differences indicated (* *p* < 0.05, ** *p* < 0.01, *** *p* < 0.001). Representative dot blot images are shown beneath each graph. Each experiment was repeated three times.

**Figure 8 bioengineering-12-01077-f008:**
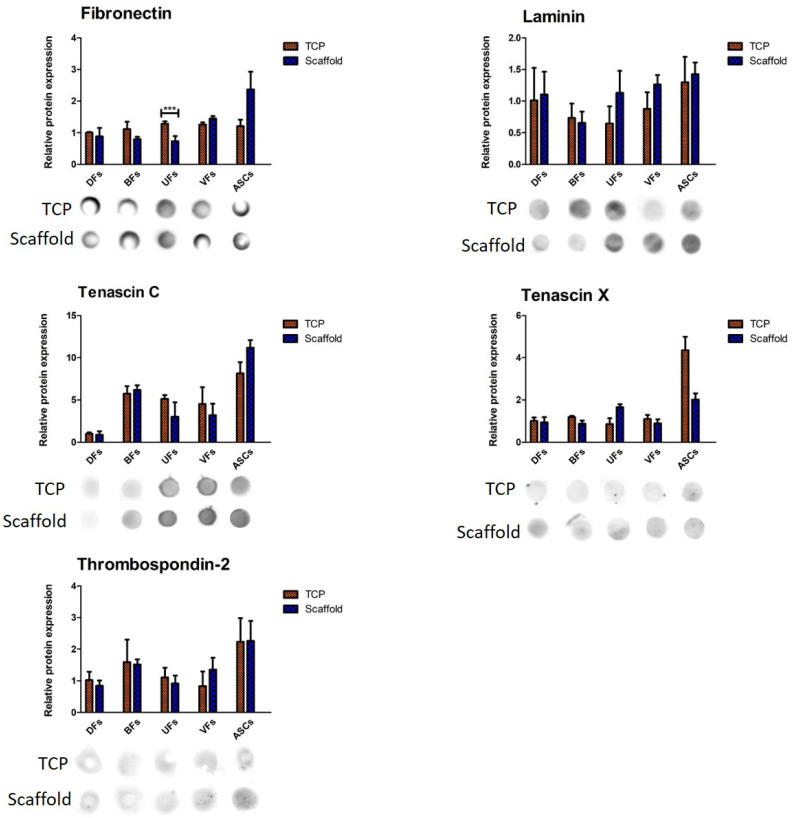
Relative expression of ECM proteins by different stromal cells cultured on PCLCOL versus TCP, assessed via dot blot analysis. Signals were detected using chemiluminescent substrate and imaged with the Fusion Fx7 system. Dot intensities were quantified using ImageJ, normalized by the culture surface area, and expressed relative to DFs cultured on TCP. Bar graphs show mean ± SD, with significant differences indicated (*** *p* < 0.001). Representative dot blot images are shown beneath each graph. Each experiment was repeated three times.

**Figure 9 bioengineering-12-01077-f009:**
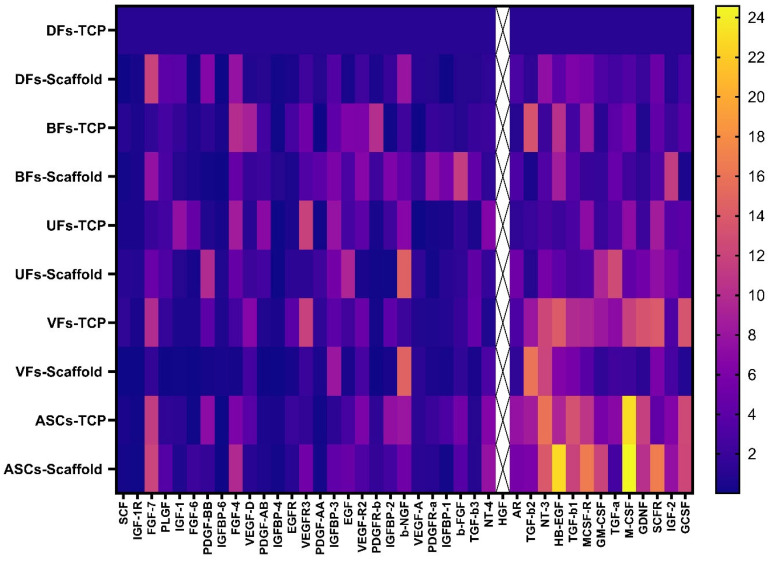
Heatmap of growth factor and cytokine secretion profiles by stromal cells cultured on TCP and PCLCOL scaffolds. The heatmap presents the relative expression levels of 41 targets secreted by DFs, BFs, UFs, VFs, and ASCs cultured on either TCP or PCLCOL. Expression values were normalized to those of DFs cultured on TCP. Color intensity reflects relative abundance, with warmer colors (yellow) representing higher expression and cooler colors (purple) representing lower expression. HGF, which showed consistently high expression across most groups, was excluded from quantification to avoid bias (marked with crossed-out column). The remaining 40 targets reveal substrate- and cell-type-specific secretion patterns, indicating that both the origin of stromal cells and the culture surface significantly influence paracrine signaling profiles.

**Figure 10 bioengineering-12-01077-f010:**
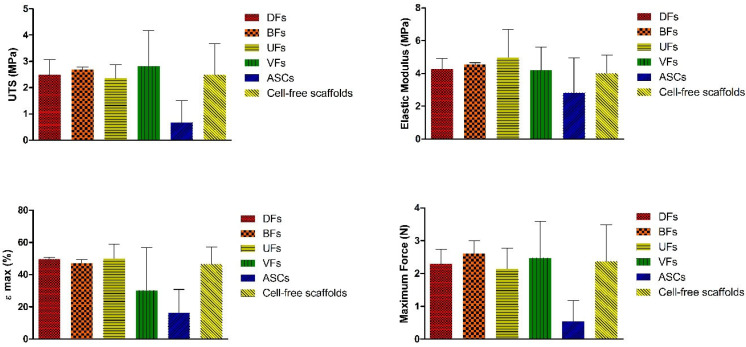
Histograms comparing the UTS, elastic modulus, εmax, and maximum force of the self-assembled ECM tissues generated on PCLCOL using different stromal cells. Each experiment was repeated three times.

**Figure 11 bioengineering-12-01077-f011:**
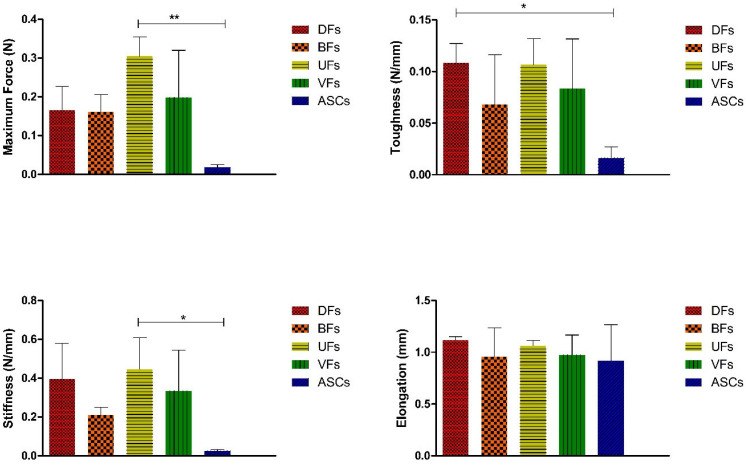
Histograms comparing the maximum force, toughness, stiffness, and elongation of the self-assembled ECM tissues generated on TCP using different stromal cells. * *p*-value < 0.05 and ** *p*-value < 0.01. Each experiment was repeated three times.

**Table 1 bioengineering-12-01077-t001:** Donor and tissue characteristics of cell sources.

Cell Type	Lab ID	Organ	Age	Sex	Ethic Approval (Laval University)
ASC	LA35	Fat (Lipoaspirate)	35	Female	DR-002-1117
BF	FHu5	Bladder	6	Male	2012-1341, DR-002-1190
DF	FD34	Skin (Dermis)	34	Female	2012-1341, DR-002-1190
UF	FU28	Urethra (proximal part)	28	Male	2012-1341, DR-002-1190
VF	FVa32	Vagina	32	Female	2012-1341, DR-002-1190

**Table 2 bioengineering-12-01077-t002:** Antibodies and their concentrations used in the dot blot assay.

Antibody	Host Species	Manufacturer	Catalogue Number	Dilution
Anti-Collagen I	Rabbit	Rockland Immunochemicals, Inc. (Burlington, VT, Canada)	600-401-105-0.1	1:5000
Anti-Collagen III	Rabbit	Cedarlane (Burlington, VT, Canada)	CL50311AP-1	1:5000
Anti-Elastin	Rabbit	Institute Pasteur, (Lyon, France)	Custom-made (lot 25011)	1:3000
Anti-Laminin	Rabbit	Abcam (Cambridge, UK)	Ab14509	1:2000
Anti-Fibronectin	Rabbit	Abcam	ab32419	1:3000
Anti-Fibrillin	Rabbit	Elastin Products Company, Inc. (Owensville, MO, USA)	PR225	1:2000
Anti- Thrombospondin 1	Mouse	Novus Biologicals (Centennial, CO, USA)	NB-100-2059	1:2000
Anti-Tenascin C	Mouse	Abcam (Cambridge, UK)	Clone BC-24	1:3000
Anti-Tenascin X	Rabbit	Santa Cruz B. (California, CA, USA)	Clone H-90	1:3000
Anti-Rabbit Horseradish Peroxidase (HRP)	Goat	Invitrogen (Waltham, MA, USA)	WC320195	1:1000
Anti-Mouse HRP	Goat	Invitrogen (Waltham, MA, USA)	WB322805A	1:1000

**Table 3 bioengineering-12-01077-t003:** Major FTIR peak assignments for PCLCOL, PCL, and Collagen scaffolds.

Sample	Wavenumber (cm^−1^)	Assignment
**PCLCOL**	~3300	Amide A
**PCLCOL**	~2940	CH_2_ asymmetric/symmetric stretching
**PCLCOL**	~1650	Amide I
**PCLCOL**	~1550	Amide II
**PCLCOL**	~1470	CH_2_ bending
**PCLCOL**	~1365	CH bending
**PCLCOL**	~1290	C–O stretching
**PCLCOL**	~1240	Amide III

## Data Availability

The raw data supporting the conclusions of this article will be made available by the authors on request.

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
