# Peer review of "Electrospun Polycaprolactone/Collagen Scaffolds Enhance Manipulability and Influence the Composition of Self-Assembled Extracellular Matrix"

_bioengineering, 2025, doi:10.3390/bioengineering12101077_

Round 1

Reviewer 1 Report

Comments and Suggestions for Authors

1. Novelty: The concept of combining PCLCOL with ECM self-assembly and further functionalization is not itself new or particularly innovative since many such hybrid scaffolds have been described. Therefore, the statement of novelty should be clearer and the unique contribution or advance of this study should be stated very clearly. 

2. Donors: There is a single donor per stromal cell type used. This makes the conclusions less generalizable and makes the results less convincing since it is well-known that MSC secretory behavior is highly donor-dependent. The authors should at least discuss this more explicitly.

3. Cell viability: The fact that cell viability initially drops on scaffolds during days 3–5 is intriguing and should have mechanistic data to back it up (what is causing it? apoptosis? reduced proliferation? metabolic dormancy?). If additional assays were not performed to clarify this, more discussion is needed to clarify the possibility of confounding factors in cell proliferation and growth factor secretion.

4. Mechanical properties: While the tensile results clearly show that the material is much more resistant to handling than fragile ECM sheets, it is not clear how this compares to native tissues. To assess whether the results are physiologically relevant, there should be a benchmark (skin, bladder, vaginal tissue, etc. ). 

5. ECM-growth factor secretion: The secretion data are extensive and impressive, but still remain descriptive. The impact on epithelialization or tissue repair has not been explored in any way. Furthermore, the rationale for excluding HGF from consideration/data presentation despite its obvious biological activity and impact needs more explanation.

Author Response

Dear reviewer,

Thanks for reviewing our manuscript and for your comments. Below are our point-by-point responses to your concerns.

  1. Novelty: The concept of combining PCLCOL with ECM self-assembly and further functionalization is not itself new or particularly innovative since many such hybrid scaffolds have been described. Therefore, the statement of novelty should be clearer and the unique contribution or advance of this study should be stated very clearly. 

Response: The statement about the novelty of this work was rewritten (Line 387-390).

“The novelty of this study lies in its systematic evaluation of tissue-specific stromal cells cultured on PCLCOL and the demonstration of how this platform distinctly modulates both ECM composition and growth factor secretion profiles compared to conventional self-assembly on TCP”.

  1. Donors: There is a single donor per stromal cell type used. This makes the conclusions less generalizable and makes the results less convincing since it is well-known that MSC secretory behavior is highly donor-dependent. The authors should at least discuss this more explicitly.

Response: This was discussed as the key limitation of this research (Line 469-475).

“A key limitation of this study is that only one donor was used per stromal cell type, which constrains the generalizability of the findings. It is well established that mesenchymal cells exhibit marked donor-to-donor variability in their secretory behavior, proliferation capacity, and ECM deposition potential, influenced by factors such as age, sex, and health status of the donor [54, 55]. Consequently, the differences in ECM composition and growth factor secretion observed here may in part reflect donor-specific characteristics rather than universal features of the tissue of origin”.

  1. Cell viability: The fact that cell viability initially drops on scaffolds during days 3–5 is intriguing and should have mechanistic data to back it up (what is causing it? apoptosis? reduced proliferation? metabolic dormancy?). If additional assays were not performed to clarify this, more discussion is needed to clarify the possibility of confounding factors in cell proliferation and growth factor secretion.

Response: We have now clarified this in the Discussion section to avoid ambiguity regarding potential confounding effects on ECM deposition and growth factor secretion (Line 414-426). 

“Despite most cells attaching to the scaffolds, an intriguing observation in this study is the transient decline in stromal cell viability on PCLCOL scaffolds during the early culture period (days 3–5). While viability recovered by day 10, the initial reduction raises questions about the underlying mechanisms. The reduced viability observed in early culture may reflect a transient dormancy phase, as stromal cells adapt to the altered mechanical and topographical cues of the scaffold. When stromal cells are cultured on scaffolds, they might enter a state of dormancy due to the distinct biophysical signals from the new microenvironment [40]. Stromal cells dormancy is characterized by a state of reduced metabolic activity. It seems that the stiffness of scaffold material has a major impact on the stromal cell’s metabolic activity [41-43]. However, further research is needed to determine whether the early decline in viability on PCLCOL scaffolds is linked to apoptosis, altered proliferation dynamics, or metabolic dormancy, and to assess how these factors may confound subsequent ECM deposition and growth factor secretion”

  1. Mechanical properties: While the tensile results clearly show that the material is much more resistant to handling than fragile ECM sheets, it is not clear how this compares to native tissues. To assess whether the results are physiologically relevant, there should be a benchmark (skin, bladder, vaginal tissue, etc.). 

Response: We fully agree that benchmarking against native tissues such as skin, bladder, or vaginal tissue would provide important context for interpreting the physiological relevance of our findings. Unfortunately, we did not have access to sufficient quantities of these native human tissues to perform comparative mechanical testing within the scope of this study. Nevertheless, we emphasize that the primary objective of this work was to evaluate whether PCLCOL scaffolds could improve the manipulability and structural robustness of self-assembled ECM constructs. Future studies will aim to incorporate direct comparisons with native tissues to more precisely situate the mechanical properties of these hybrid scaffolds within a physiological framework.

  1. ECM-growth factor secretion: The secretion data are extensive and impressive, but still remain descriptive. The impact on epithelialization or tissue repair has not been explored in any way. Furthermore, the rationale for excluding HGF from consideration/data presentation despite its obvious biological activity and impact needs more explanation.

Response: We agree that establishing the functional impact of the observed ECM and growth factor secretion profiles on epithelialization and tissue repair would significantly strengthen the translational relevance of this work. However, such an investigation would require an additional set of experiments, including epithelial cell co-cultures or in vivo models, which were beyond the scope of the present study. We have highlighted this as an important direction for future research in the revised conclusion (purple highlight). Regarding HGF, although HGF is biologically active and relevant to tissue repair, it was consistently and strongly overexpressed across nearly all groups. This high baseline abundance introduced bias by masking the relative differences in secretion profiles of other growth factors, making comparisons difficult. For this reason, HGF was excluded from the quantitative analysis (line 344-348 and 492-495).

“Hepatocyte Growth Factor (HGF) is a biologically active and relevant to tissue repair [23], that was consistently and strongly overexpressed across nearly all groups. This high baseline abundance introduced bias by masking the relative differences in secretion profiles of other growth factors. For this reason, HGF was excluded from the quantitative analysis.”.

” Future studies will focus on directly assessing the functional consequences of the altered ECM composition and growth factor secretion, particularly their impact on epithelialization and tissue repair, through epithelial co-culture models and in vivo validation”.

Reviewer 2 Report

Comments and Suggestions for Authors

This work addresses the use of polycaproplactone-collagen scaffolds for culture of connective tissue cells. Scaffolds should provide structural support to the cells and enhance deposit of extracellular matrix by cells. By this way, the mechanic properties of the artificial scaffold is combined with the de novo secreted matrix of connective tissue cells to provide an optimized substrate for down-stream application. The topic is interesting and relevant. The study is not entirely convincing and some additional work has to be done prior to publication.

“In this regard, Carrier et al. investi-48 gated how the tissue origin of stromal fibroblasts and epithelial cells influenced corneal 49 reconstruction in vitro. Using self-assembled human tissue models, they combined cor-50 neal or skin-derived fibroblasts and epithelial cells in various pairings. They found that 51 the differentiation, epithelial thickness, and transparency of the reconstructed tissues 52 were significantly affected by the organ-specific origin of the cells.”

Comment: This is an important point.  In this sentence, fibroblasts are discussed. In the previous sentence it is about MSCs. These cells are not the same, please do not mix MSCs and fibroblasts in the same paragraph in a way that suggests it is more or less the same. Can you please explain in the introduction more clearly, how these cells are distinguished and which effects fibroblasts or MSCs have in organizing epithelia for example.

“2-8. Cell viability assay 150

Scaffolds were cut, sterilized and prepared for cell seeding according to the protocol 151 as described in section 2-8.”

Comment: I think you want to refer to section 2-7, not 2-8 ?

“2-13. Growth factors secretion profile analysis 213

The scaffolds were cut, sterilized, and prepared for cell seeding according to the pro-214 tocol described in section 2-8.”

Comment: Also here is assume you want to refer to 2-7, not 2-8?

“xxAdd a sen-239 tence on size distribution, it is important results…(in C), why no numbering of each 240 panel?xx 241”

Comment: Please finalize the text. And your PI is right. The numbering of the panels should be added and described in the figure legend. Frequency is shown in %, I assume. Pls add that to the axis.

“DFs, BFs, UFs, and VFs, and ASCs were obtained from healthy human donors undergoing surgery for a non-oncologic condition, and utilized for cell culture studies [17].”

Comment: Authors provide a 18 years old manuscript as reference. The MS deals with adipogenic stromal cells. It has nothing to do with the other cell types. This is not an appropriate way of citing the literature.

Further comment

  • There is zero characterization of the cells shown in this manuscript. A basic characterization of the biomaterial is followed by seeding experiments. The identity of the different cells types is nothing more than a claim. To become acceptable for publication a basic characterization of the cells is a minimum requirement.

“Comparing the ECM thickness between groups, we found that UFs, VFs, and ASCs cul- tured on PCLCOL produced significantly thicker ECM than the same cells cultured on TCP. However, the thickness of ECM produced by DFs and BFs cultured on TCP and 289

PCLCOL was not significantly different.”

Comment: This can be due to differences in cell proliferation leading to different numbers of cell layers. It is necessary to analyse the cell layers for each sample to conclude if the cells have grown in one or multiple layers. This has to be done on different samples from the engineered tissue, not on a single slide. For the conclusion it is important to know if there are differences in the number of cell layers between the groups.

Further comments

  • Quantification of dot plot signals is not fully convincing. For example VFs on TCP show a substantial darker spot for Fibrillin as compared to Scaffold but are quantified as same. You may argue that the dots have different size that compensate for the signal. But why have the dots different size? The drops that are placed on the membrane should be the same.
  • How do you exclude that collagen released from the scaffold contaminates the samples for protein measurement? To my mind, cell-free scaffold needs to be tested in addition for collagen detection with the dot blot assay.

Author Response

Dear reviewer 2,

Thanks for reviewing our manuscript and for your comments. Below are our point-by-point responses to your concerns.

This work addresses the use of polycaproplactone-collagen scaffolds for culture of connective tissue cells. Scaffolds should provide structural support to the cells and enhance deposit of extracellular matrix by cells. By this way, the mechanic properties of the artificial scaffold is combined with the de novo secreted matrix of connective tissue cells to provide an optimized substrate for down-stream application. The topic is interesting and relevant. The study is not entirely convincing and some additional work has to be done prior to publication.

“In this regard, Carrier et al. investigated how the tissue origin of stromal fibroblasts and epithelial cells influenced corneal reconstruction in vitro. Using self-assembled human tissue models, they combined corneal or skin-derived fibroblasts and epithelial cells in various pairings. They found that the differentiation, epithelial thickness, and transparency of the reconstructed tissues were significantly affected by the organ-specific origin of the cells.”

Comment: This is an important point.  In this sentence, fibroblasts are discussed. In the previous sentence it is about MSCs. These cells are not the same, please do not mix MSCs and fibroblasts in the same paragraph in a way that suggests it is more or less the same. Can you please explain in the introduction more clearly, how these cells are distinguished and which effects fibroblasts or MSCs have in organizing epithelia for example.

Response: The distinction between MSCs and fibroblast cells was highlighted in the introduction section (Line 51-57).

“It is important to emphasize the distinction between mesenchymal stem/stromal cells (MSCs) and fibroblasts. MSCs are multipotent progenitor cells capable of differentiating into multiple mesenchymal lineages and are characterized by a broad secretory profile that supports epithelial growth, repair, and immune modulation. In contrast, fibroblasts represent a more lineage-committed stromal cell population, primarily responsible for producing and remodeling ECM within their tissue of origin.”

“2-8. Cell viability assay

Scaffolds were cut, sterilized and prepared for cell seeding according to the protocol as described in section 2-8.”

Comment: I think you want to refer to section 2-7, not 2-8?

Response: We corrected this error. Thank you!

“2-13. Growth factors secretion profile analysis

The scaffolds were cut, sterilized, and prepared for cell seeding according to the protocol described in section 2-8.”

Comment: Also here is assume you want to refer to 2-7, not 2-8?

Response: We corrected this error.

“Add a sentence on size distribution, it is important results…(in C), why no numbering of each panel?”

Comment: Please finalize the text. The numbering of the panels should be added and described in the figure legend. Frequency is shown in %, I assume. Pls add that to the axis.

Response: The panels of figure 1 were labelled and the Y axis was changed from the value numbers to percentage of frequency.

“DFs, BFs, UFs, and VFs, and ASCs were obtained from healthy human donors undergoing surgery for a non-oncologic condition, and utilized for cell culture studies [17].”

Comment: Authors provide a 18 years old manuscript as reference. The MS deals with adipogenic stromal cells. It has nothing to do with the other cell types. This is not an appropriate way of citing the literature.

Further comment

  • There is no characterization of the cells shown in this manuscript. A basic characterization of the biomaterial is followed by seeding experiments. The identity of the different cells types is nothing more than a claim. To become acceptable for publication a basic characterization of the cells is a minimum requirement.

Response: We realize the oversight of not providing all of the references needed to better identify the cells used and previously characterized.

DFs were evaluated for morphology, size, proliferation/doubling time, and metabolic activity (https://doi.org/10.3390/ijms231710035).

BFs were evaluated by morphology and immunofluorescence marker expression (https://doi.org/10.5489/cuaj.2953).

The characterization data for UFs has not been published yet.

VFs were evaluated for morphology, marker expression, and proliferative/ECM-secreting capacity across passages (https://doi.org/10.1016/j.trsl.2016.07.019).

ASCs were evaluated for isolation yield, proliferation capacity, adipogenic differentiation, ECM secretion, and functional output (https://doi.org/10.1016/j.biomaterials.2007.02.030).

 Citations were added to support this characterization (line 130).

 “Comparing the ECM thickness between groups, we found that UFs, VFs, and ASCs cultured on PCLCOL produced significantly thicker ECM than the same cells cultured on TCP. However, the thickness of ECM produced by DFs and BFs cultured on TCP and PCLCOL was not significantly different.”

Comment: This can be due to differences in cell proliferation leading to different numbers of cell layers. It is necessary to analyse the cell layers for each sample to conclude if the cells have grown in one or multiple layers. This has to be done on different samples from the engineered tissue, not on a single slide. For the conclusion it is important to know if there are differences in the number of cell layers between the groups.

Response: In our study, we used the same seeding protocol and density for all stromal cell types, with two sequential seeding steps at full density, to minimize variability related to initial cell number. This approach was applied consistently across groups to ensure that any differences observed in ECM thickness were attributable to scaffold–cell interactions rather than disparities in seeding strategy. We acknowledge that analyzing the number of cell layers across multiple independent sections would provide additional mechanistic insight into whether ECM thickness correlates with multilayer growth. However, such an analysis was not feasible within the current study, as it would require a separate series of experiments designed specifically to quantify stratification. Instead, our primary focus was to compare ECM deposition between substrates under identical seeding conditions.

 Further comments

Quantification of dot plot signals is not fully convincing. For example VFs on TCP show a substantial darker spot for Fibrillin as compared to Scaffold but are quantified as same. You may argue that the dots have different size that compensate for the signal. But why have the dots different size? The drops that are placed on the membrane should be the same.

Response: To avoid the discrepancy between visual inspection and quantified data in Figure 7, we have updated the figure panels to present representative blot images corresponding to the averaged quantification results. Since the dot blot experiments were performed in triplicate, we selected representative replicates that best reflected the quantified densitometry values, ensuring consistency between the images and the data shown in the graphs.

  • How do you exclude that collagen released from the scaffold contaminates the samples for protein measurement? To my mind, cell-free scaffold needs to be tested in addition for collagen detection with the dot blot assay.

Response: To prevent any contribution from the collagen embedded in the PCLCOL scaffold, we probed collagen with human-specific monoclonal antibodies, so the assay selectively detects human collagen synthesized by the stromal cells and does not cross-react with the rat-derived collagen present in the scaffold.

Reviewer 3 Report

Comments and Suggestions for Authors

this study demonstrates that electrospun PCLCOL provide a mechani-
cally robust and biologically responsive platform for stromal cell culture. These scaffolds
support tissue-specific ECM deposition and modulate growth factor secretion. Some improvements can be done 

  1. I suggest to include a scheme that describes the entire idea of the study.
  2. The IR data presentation must be improved , a table with all peaks assignments is required. https://pubs.acs.org/doi/10.1021/acsabm.3c01108
  3. The indirect adhesion assay should be replaced with a direct adhesion assay as described in https://pubs.acs.org/doi/10.1021/acsabm.3c01108 
  4. The mechanical data should be analysed in a statistical manner , ROC analyses can be included

Author Response

Dear reviewer,

Thanks for reviewing our manuscript and for your comments. Below are our point-by-point responses to your concerns.

This study demonstrates that electrospun PCLCOL provide a mechanically robust and biologically responsive platform for stromal cell culture. These scaffolds support tissue-specific ECM deposition and modulate growth factor secretion. Some improvements can be done 

  1. I suggest to include a scheme that describes the entire idea of the study.

Response: We added the graphical abstract that describes the entire idea of the whole study.

  1. The IR data presentation must be improved, a table with all peaks assignments is required. https://pubs.acs.org/doi/10.1021/acsabm.3c01108

Response: A new table (table 3) was included that describes the important peaks of the composite scaffold.

Sample

Wavenumber (cm⁻¹)

Assignment

PCLCOL

~3300

Amide A

PCLCOL

~2940

CH₂ asymmetric/symmetric stretching

PCLCOL

~1650

Amide I

PCLCOL

~1550

Amide II

PCLCOL

~1470

CH₂ bending

PCLCOL

~1365

CH bending

PCLCOL

~1290

C–O stretching

PCLCOL

~1240

Amide III

  1. The indirect adhesion assay should be replaced with a direct adhesion assay as described in https://pubs.acs.org/doi/10.1021/acsabm.3c01108 

Response: We agree that a direct adhesion assay, as described in the cited reference, would provide more precise quantitative information. However, due to limitations in resources and equipment, we were unable to implement this approach in the present study.

  1. The mechanical data should be analysed in a statistical manner, ROC analyses can be included

Response: Because two completely different methods were required to assess the mechanical properties of the hybrid constructs (uniaxial tensile testing) and the ECM sheets generated on TCP (biaxial puncture test), the resulting datasets are not directly comparable. For this reason, statistical comparisons such as ROC analysis could not be applied across the two groups. Instead, our intention was to demonstrate the relative manipulability and robustness of the hybrid scaffolds, rather than to equate their absolute mechanical values with those of the fragile ECM sheets.

Round 2

Reviewer 2 Report

Comments and Suggestions for Authors

Most of my concerns were addressed in the revised version of the manuscript.

Reviewer 3 Report

Comments and Suggestions for Authors

The revision is okay to me